# Anticancer Peptides Derived from Aldolase A and Induced Tumor-Suppressing Cells Inhibit Pancreatic Ductal Adenocarcinoma Cells

**DOI:** 10.3390/pharmaceutics15102447

**Published:** 2023-10-11

**Authors:** Changpeng Cui, Qingji Huo, Xue Xiong, Kexin Li, Melissa L. Fishel, Baiyan Li, Hiroki Yokota

**Affiliations:** 1Department of Pharmacology, School of Pharmacy, Harbin Medical University, Harbin 150081, China; cuich@iu.edu (C.C.); qinghuo@iu.edu (Q.H.); xiongxue@iu.edu (X.X.); kexinli0104@gmail.com (K.L.); 2Department of Biomedical Engineering, Indiana University Purdue University Indianapolis, Indianapolis, IN 46202, USA; 3Department of Pediatrics, Wells Center for Pediatric Research, Indiana University School of Medicine, Indianapolis, IN 46202, USA; mfishel@iu.edu; 4Department of Pharmacology and Toxicology, Indiana University School of Medicine, Indianapolis, IN 46202, USA; 5Indiana University Simon Comprehensive Cancer Center, Indianapolis, IN 46202, USA; 6Department of Pediatrics, Indiana Center for Musculoskeletal Health, Indiana University School of Medicine, Indianapolis, IN 46202, USA

**Keywords:** pancreatic ductal carcinoma, peptide, ALDOA, induced tumor-suppressing cells

## Abstract

PDAC (pancreatic ductal adenocarcinoma) is a highly aggressive malignant tumor. We have previously developed induced tumor-suppressing cells (iTSCs) that secrete a group of tumor-suppressing proteins. Here, we examined a unique procedure to identify anticancer peptides (ACPs), using trypsin-digested iTSCs-derived protein fragments. Among the 10 ACP candidates, P04 (IGEHTPSALAIMENANVLAR) presented the most efficient anti-PDAC activities. P04 was derived from aldolase A (ALDOA), a glycolytic enzyme. Extracellular ALDOA, as well as P04, was predicted to interact with epidermal growth factor receptor (EGFR), and P04 downregulated oncoproteins such as Snail and Src. Importantly, P04 has no inhibitory effect on mesenchymal stem cells (MSCs). We also generated iTSCs by overexpressing ALDOA in MSCs and peripheral blood mononuclear cells (PBMCs). iTSC-derived conditioned medium (CM) inhibited the progression of PDAC cells as well as PDAC tissue fragments. The inhibitory effect of P04 was additive to that of CM and chemotherapeutic drugs such as 5-Flu and gemcitabine. Notably, applying mechanical vibration to PBMCs elevated ALDOA and converted PBMCs into iTSCs. Collectively, this study presented a unique procedure for selecting anticancer P04 from ALDOA in an iTSCs-derived proteome for the treatment of PDAC.

## 1. Introduction

Pancreatic ductal adenocarcinoma (PDAC) stands as the most prevalent and aggressive form of pancreatic cancer, constituting approximately 85 to 90% of all instances [1]. Nestled deep within the abdominal cavity, the pancreas poses a challenge for early PDAC detection. Initial stages of PDAC often remain inconspicuous, and when symptoms do manifest, they frequently signal an advanced tumor beyond the point of surgical removal [2,3]. Surgical resection offers the greatest efficacy for those with early-diagnosed PDAC [4]. To address PDAC at this stage, FOLFIRINOX, an adjuvant chemotherapy regimen, incorporating fluorouracil (5-Flu), leucovorin, irinotecan, and oxaliplatin, is the recommended course of action [5,6]. Nonetheless, PDAC frequently develops resistance to chemotherapy [7]. While radiation treatment can mitigate pain, its ability to impede PDAC progression often proves ineffective [8].

Due to their distinct manufacturing process and superior quality control when compared to other cell and protein-based agents, anticancer peptides (ACPs) are emerging as highly promising candidates for cancer treatment. These peptides typically consist of basic and hydrophobic residues, carrying positive charges [9], which aligns with the negatively charged nature of most cancer cells [10]. Despite the availability of various databases such as CancerPPD [11], ImmunoPeptidome [12,13], and PeptideCutter [14], which aid in the design of ACP candidates and the prediction of their tumor-binding epitopes [15], the effective design of ACPs remains a formidable challenge. Specifically, identifying precise targets and discerning cancer cells from normal cells continue to pose significant obstacles.

For the treatment of PDAC, we selected and tested ten ACP candidates that were derived from tumor-suppressing proteins, which were secreted from induced tumor-suppressing cells (iTSCs) [16,17,18,19,20]. We have previously shown that iTSCs and their tumor-suppressive conditioned medium (CM) can be generated from cancer and non-cancer cells by modulating varying signaling pathways. Interestingly, iTSCs can be generated by the overexpression of cMyc or Oct4 [17], whereas the generation of induced pluripotent stem cells (iPSCs) requires the overexpression of four transcription factors, such as cMyc, Oct4, Klf4, and Sox2. Our previous findings also demonstrated that the CM of mesenchymal stem cells (MSCs) overexpressing Akt and β-catenin selectively inhibited tumor cell proliferation and invasion [21]. Furthermore, the activation of PKA signaling in Jurkat T lymphocytes or inhibition of AMPK signaling in human peripheral blood mononuclear cells (PBMCs) were shown effective in generating iTSCs [22]. Notably, the proteomic analysis showed that CM was enriched with proteins such as enolase 1, ubiquitin C, calreticulin, and aldolase A (ALDOA), which may act as both pro- and anticancer proteins depending on their intracellular and extracellular locations [23].

A distinctive aspect of this study lies in the unique selection of ten prospective ACP candidates (P01 to P10) derived exclusively from trypsin-digested proteomes of iTSCs-derived conditioned media (CM). Following trypsin digestion of iTSC CM, we utilized mass spectrometry analysis to identify these peptides exhibiting notably high expression levels. Since trypsin typically cleaves lysine and arginine residues [24], the selected peptides all contain either lysine or arginine residues in the carboxyl end. In a cell-viability prescreening process, P04 showed the most promising anticancer capabilities and thus this study mostly focused on P04. P04 was derived from ALDOA, which is an enzyme involved in a glycolytic pathway. ALDOA is highly expressed in cancer tissues, and its elevated expression level usually indicates a poor prognosis [25]. Paradoxically, however, extracellular ALDOA acts as a tumor-suppressing protein in breast cancer [23]. We investigated in this study whether ALDOA-derived P04 is a potential candidate for the treatment of PDAC, and whether it may act additively with chemotherapeutic agents such as 5-Flu and gemcitabine (Gem).

EGFR is overexpressed or mutated in a variety of cancer types, making it an important target for anticancer therapy [26,27]. For instance, an artificially designed ACP, Disruptin, is reported to inhibit the binding of heat shock protein 90 to EGFR [28]. Similarly, this study showed that ALDOA-derived P04 can inhibit K-Ras pathways by its potential interaction with EGFR. In addition to the utilization of P04, our approach encompassed the generation of iTSCs and the production of tumor-suppressive CM via two distinct methodologies: ALDOA overexpression and the application of low-intensity vibrations (LIV). Similar to Akt within the PI3K signaling cascade and β-catenin within the Wnt signaling pathway, we conjectured that the activation of oncogenic ALDOA within the intracellular domain could potentially yield iTSCs and engender the creation of tumor-suppressing CM. Furthermore, we postulated that mechanical stimuli, like LIV, could potentially prompt the generation of iTSCs by inciting the activation of oncogenic PI3K signaling.

## 2. Materials and Methods

### 2.1. Cell Culture

Human pancreatic ductal adenocarcinoma (PDAC) cell lines including PANC1 (ATCC, Manassas, VA, USA), PANC198, and Pa03C (sourced from Indiana University Simon Comprehensive Cancer Center, Indianapolis, IN, USA) were cultured in DMEM (10-013-CV, Corning Inc., Corning, NY, USA). The breast cancer cell lines MDA-MB-231 and MDA-MB-436 (ATCC) were grown in RPMI-1640 medium (Gibco, Carlsbad, CA, USA). The culture media for all cell lines were supplemented with 10% fetal bovine serum and antibiotics (100 units/mL penicillin and 100 µg/mL streptomycin; Life Technologies, Grand Island, NY, USA). Jurkat T lymphocytes [29] (ATCC) were cultured in RPMI-1640 medium (10-040-CV, Corning). Murine MSCs derived from the bone marrow of C57BL/6 mice were cultured in DMEM. All cell cultures were maintained at 37 °C with 5% CO_2_. The overexpression of ALDOA was accomplished by transfecting its plasmids (109,854; Addgene, Cambridge, MA, USA), while blank plasmids (FLAG-HA-pcDNA3.1; Addgene) were employed as controls.

PDAC cells were subjected to treatment with ten peptides (P01 to P10, Genscript Biotech, Piscataway, NJ, USA), along with chemotherapeutic agents like gemcitabine (3259, Tocris Bioscience, Minneapolis, MN, USA) and 5-fluorouracil (3257, Tocris Bioscience). Additionally, PDAC cells received treatment with human recombinant ALDOA protein (MBS8248528, MyBioSource, San Diego, CA, USA). For in vitro experiments, conditioned medium (CM) was prepared using a low-speed centrifugation step at 2000 rpm for 10 min. The resulting supernatants were subjected to further centrifugation at 4000 rpm for 10 min and subsequently filtered using a 0.22 μm polyethersulfone membrane (Sigma, St. Louis, MO, USA). This process ensured the removal of particulate matter and debris from the CM.

### 2.2. MTT Assay and EdU Assay

MTT-based assessment of metabolic activity involved seeding approximately 2000 cells into 96-well plates. These cells were subsequently incubated with treatment agents for two days. On day 4, the cells were subjected to staining with 0.5 mg/mL thiazolyl blue tetrazolium bromide (M5655, Sigma) [30]. The determination of metabolic activity was based on the measurement of optical density at 562 nm. For the evaluation of proliferation activity using the EdU assay, around 1000 cells were seeded into 96-well plates. A fluorescence-based cell proliferation kit (Click-iT™ EdU Alexa Fluor™ 488 Imaging Kit; Thermo Fisher, Waltham, MA, USA) was employed, following the manufacturer’s protocol [31]. This kit facilitated the detection of EdU incorporation as an indicator of proliferating cells.

### 2.3. Scratch Assay

To analyze the two-dimensional migratory behavior of PANC cells, a wound-healing scratch assay was employed to assess the change in the scratched area. Around 3 × 10^5^ cells were seeded into 12-well plates on day 1. On day 2, a scratch was carefully created on the cell layer using the tip of a plastic pipette [32]. The cells were then rinsed with DMEM to eliminate any detached cells. Images of the cell-free areas were captured at both 0 h and 24 h using a light microscope (40× magnification). The sizes of the cell-free zones were subsequently quantified using Image J (v1.5.4f).

### 2.4. Transwell Invasion Assay

To assess the invasive potential of tumor cells, a 12-well plate and transwell chambers (353,182, Thermo Fisher) featuring 8-µm pore size were employed. The transwell chambers were pre-coated with 300 µL of Matrigel (100 µg/mL), followed by the addition of 500 µL of serum-free medium. Afterward, the chambers were rinsed three times with a serum-free medium. Subsequently, approximately 7 × 10^4^ cells in 300 µL of serum-free DMEM were introduced into the upper chamber, while 800 µL of the conditioned medium (CM) was added to the lower chamber [33]. Upon completion of the experiment, cells that had invaded the lower side of the membrane were fixed and stained using methanol and crystal violet. Utilizing an inverted optical microscope (100×), five randomly selected images were captured. From these images, the average count of stained cells, serving as an indicator of the invasion capacity, was determined.

### 2.5. Western Blot Analysis

Cell lysis was achieved using RIPA lysis buffer (sc-24948, Santa Cruz Biotech, Dallas, TX, USA), supplemented with protease inhibitors (PIA32963, Thermo Fisher), and phosphatase inhibitors (2006643, Calbiochem, Billerica, MA, USA). Subsequently, proteins were separated using 10–15% SDS gels (Bio-Rad Laboratories, Hercules, CA, USA) and then transferred onto polyvinylidene difluoride membranes (IPVH00010, Millipore, Billerica, MA, USA). The membrane was first incubated with primary antibodies, followed by secondary antibodies (7074S/7076S, Cell Signaling, Danvers, MA, USA). Antibodies against ALDOA, Snail, p-Src, Src, cleaved caspase 3, caspase 3 (Cell Signaling), and K-Ras (Santa Cruz Biotech) were employed, with β-actin (Sigma) serving as a control. Protein levels were determined utilizing the Super-Signal West Femto Maximum Sensitivity Substrate (PI34096, Thermo Fisher), and signal intensities were quantified using a luminescent image analyzer (LAS-3000, Fuji Film, Tokyo, Japan) [34].

### 2.6. Human Peripheral Blood Mononuclear Cells (PBMCs)

Human peripheral blood samples were obtained following the principles outlined in the Declaration of Helsinki. The Ethics Committee of Osaka University approved the use of these blood samples under protocol #21344. A total of 8 mL of peripheral blood was collected from 10 healthy volunteers with an average age of 34.2 years, ranging from 23 to 54 years. To each blood sample, an equal volume of 0.9% NaCl solution was added, after which 6 mL of the mixture was combined with 3 mL of lymph prep solution (1,114,544, Abbott Diagnostics Technologies AS, Oslo, Norway). This mixture was then subjected to centrifugation at 800× *g* for 30 min at room temperature. Following centrifugation, peripheral blood mononuclear cells (PBMCs) were meticulously collected using a Pasteur pipette, and these cells were subsequently cultured in AlyS705 medium (Cell Science & Technology Institute, Inc., Sendai, Japan).

### 2.7. Ex Vivo Tissue Assay

The utilization of human pancreatic cancer tissue was approved by the Indiana University Institutional Review Board (1911155674), and the tissue samples were sourced from the Indiana University Simon Comprehensive Cancer Center Tissue Procurement Core. A freshly procured pancreatic cancer tissue, weighing approximately 1 g, was manually sectioned into small fragments (ranging from 0.5 to 0.8 mm in length) using a scalpel. These fragments were subsequently cultured in DMEM supplemented with 10% FBS and antibiotics for one day. Following this initial incubation, the described conditioned medium (CM) was introduced and maintained for an additional four days, during which any alterations in fragment size were assessed.

### 2.8. Molecular Docking Analysis

For the assessment of interactions involving epidermal growth factor receptor (EGFR) and ALDOA, as well as EGFR and P05, we utilized the ZDOCK program (version 2016, Discovery Studio, San Diego, CA, USA) [35]. The structures of EGFR and ALDOA were sourced from the PDB database (EGFR PDB ID: 1mox, ALDOA PDB ID: 7u5). P04 was derived from the ALDOA sequences. Following the removal of solvent and the deletion of unnecessary atoms, we conducted ZDOCK analysis to explore all potential binding orientations within the translational and rotational space, linking EGFR with either ALDOA or P04. Each orientation was then subjected to evaluation through the energy-based ZDOCK scoring system [36]. Furthermore, we employed the protein–protein interaction module in Discovery Studio software 2016 and predicted potential hydrogen-bonding interactions in both the EGFR-ALDOA and EGFR-P04 complexes. To validate the docking method, we utilized interactions between EGFR and transforming growth factor alpha (TGFA) as a positive control.

### 2.9. Vibration Assay

For the generation of an anti-PDAC conditioned medium (CM), a vertical low-intensity vibration (LIV) at a frequency of 90 Hz and 2 g-level was applied to Jurkat cells and PBMCs (approximately 3 × 10^5^ cells) for two sessions lasting 20 min each, separated by a 3 h interval. After the second LIV session, the culture medium was replaced with a fresh medium. The cells were then incubated for 24 h, during which the CM was collected. To increase the viscosity of the culture medium from 0.8 cP to 20 cP, a solution of 0.1% methylcellulose (#428430500, Thermo Fisher) was introduced into the medium.

### 2.10. Statistical Analyses

For cell-based investigations, we conducted three to four distinct experiments, with results presented as mean ± standard deviation (S.D.). To assess statistical significance, we employed a one-way analysis of variance (ANOVA). Following this, post hoc comparisons with control groups were executed using Bonferroni correction, with statistical significance set at *p* < 0.05. In the figures, single and double asterisks denote *p* < 0.05 and *p* < 0.01, respectively.

## 3. Results

### 3.1. Anti-Tumor Effect of P02, P04, and P07 on PDAC Cells

Initially, we conducted an evaluation of ten prospective anti-tumor peptides, designated as P01 to P10, sourced from the CM obtained from MSCs. Based on our previous studies [17], these MSCs were engineered to express anti-tumor proteins (Appendix A). The peptides selected for evaluation were derived from trypsin-digested fragments encompassing all proteins present within CM. These peptides were assessed for their potential to impede cancer growth, using three distinct PDAC cell lines: PANC1, PAN198, and PA03C. Among the array of peptides under scrutiny, namely P02, P04, and P07, these specific peptides exhibited a notable average reduction in cell viability (Appendix A). Their corresponding amino acid sequences were enumerated (Figure 1A). Remarkably, when evaluating the trio of peptides, P04 stood out as manifesting the most potent anti-tumor effects, particularly evident in its capacity to curtail MTT-based cell viability within PANC1 cells (Figure 1B). Subsequent to this observation, our focus extended to examining the inhibitory potentials of these three peptides concerning both cell viability and cell motility. The findings from our scratch assay revealed that P02, P04, and P07 each contributed to the reduction in the motility of PANC1 cells, with P04 displaying the most pronounced anti-motility impact (Figure 1C). Consequently, our attention converged on P04, probing its influence on the proliferation and invasion tendencies of pancreatic cancer cells. Employing the EdU assay and the transwell invasion assay, we demonstrated that P04 exhibited a discernible decrease in the proliferation and invasive characteristics of PANC1 cells (Figure 1D,E).

### 3.2. Anti-Tumor Effect of P04 on Human PDAC Tissue Fragments

To further assess the potency of P04 as an anti-tumor agent, we subjected freshly obtained human pancreatic cancer tissue fragments to P04 treatment. The experimental setup involved the acquisition of PDAC tissue from a patient, which was subsequently sectioned into smaller fragments. These fragments were then divided into two groups: one cultured in a standard medium and the other treated with a concentration of 25 μg/mL of P04. Over a span of four days, we closely monitored the alterations in their morphology at 48 h intervals (Figure 2A). The outcomes gleaned from this investigation showcased a notable reduction in the size of ex vivo PDAC fragments over the course of the four-day treatment with P04 (Figure 2B,C). Furthermore, our observations indicated that the combined application of P04 and 5-Flu exhibited superior efficacy compared to the usage of 5-Flu alone, resulting in a more substantial reduction in the size of PDAC fragments (Figure 2D,E).

### 3.3. Enhanced Anti-Tumor Effects in Combination with Chemotherapeutic Agents

We proceeded to delve into the synergistic impacts achieved by combining P04 with two commonly utilized chemotherapeutic agents, namely gemcitabine and 5-Flu. The introduction of P04 at a concentration of 25 μg/mL yielded a discernible reduction in the IC50 value of gemcitabine, shifting it from 1.0 μM to 0.8 μM, and similarly, that of 5-Flu decreased from 3.3 μM to 1.6 μM, as evidenced in the MTT-based viability assay of PANC1 cells (Figure 3A). Additionally, the administration of P04 elicited a downregulation of oncogenic proteins, including p-Src and Snail, within both PANC1 and PAN198 cells. Moreover, P04 exhibited its cytotoxic potential by inducing an elevation in the levels of cleaved caspase 3, a recognized marker of apoptosis (Figure 3B).

### 3.4. Generation of Antitumor CM by the Overexpression of ALDOA in MSCs

ALDOA operates as a cytoplasmic enzyme that facilitates the reversible conversion of fructose-1,6-bisphosphate into glyceraldehyde 3-phosphate and dihydroxyacetone phosphate within the context of glycolysis. A comprehensive analysis of the TCGA database unveiled an association between heightened transcript levels of Aldoa in PDAC tissues and a compromised survival rate (Figure 4A,B). Building upon our preceding investigations, wherein several tumor-suppressive proteins, including Moesin and Enolase 1, were enriched in tumor-suppressive CM, prompting the conversion of both tumor and non-tumor cells into iTSCs, we turned our focus to the role of ALDOA. In this study, we orchestrated the overexpression of ALDOA in MSCs, subsequently assessing the therapeutic potential of their corresponding CM. Intriguingly, the CM derived from ALDOA-overexpressing MSCs demonstrated a profound tumor-suppressive effect by curtailing the viability, proliferation, motility, and invasion of PANC1 cells (Figure 4C–E), along with PANC198 cells (Appendix A).

The utilization of recombinant human ALDOA protein (rhALDOA) exhibited a notable reduction in MTT-based cell viability (Figure 5A). Additionally, the outcomes from Western blotting revealed discernible decreases in both ALDOA and K-Ras levels following the administration of rhALDOA (Figure 5B). Our previous investigations underscored that the anti-tumor efficacy of ALDOA is facilitated by its interaction with EGFR. In this study, a protein docking analysis yielded predictions that P04, a peptide fragment derived from ALDOA, might exhibit binding with EGFR through four potential hydrogen bond interactions within the range of 2.68 to 3.07 Å (Figure 5C, Appendix A).

### 3.5. Generation of Tumor-Suppressive CM from PBMCs by Overexpression ALDOA

Subsequently, we investigated the possibility of generating anti-PDAC CM from patient-derived PBMCs through the process of ALDOA overexpression. Notably, the levels of ALDOA were observed to increase following the transfection of ALDOA plasmids (Figure 6A). Our findings unveiled that CM obtained from PBMCs led to a reduction in the MTT-based viability of PANC1 and PAN198 cells (Figure 6B). Furthermore, the anti-tumor potential of PBMC-derived CM was validated using patient-derived PDAC tissue fragments. Remarkably, the inhibitory impact of PBMC-derived CM displayed an additive effect alongside that of P04 (Figure 6C,D).

### 3.6. Generation of Tumor-Suppressive CM Using Low-Intensity Vibration

Finally, we explored the feasibility of generating anti-PANC CM from PBMCs through the application of LIV at a frequency of 90 Hz with a force level of 2 g. The results demonstrated that CM obtained from LIV-treated lymphocytes exhibited a reduction in the MTT-based viability of PANC1 cells (Figure 7A). Western blot analysis unveiled an elevation in ALDOA levels in PBMCs following LIV treatment (Figure 7B), and their corresponding CM led to diminished MTT-based viability and restrained scratch-based motility of PANC1 cells (Figure 7C). Critically, the CM derived from PBMCs did not exert adverse effects on the viability and motility of MSC cells (Figure 7D). The efficacy of LIV-treated CM in exerting its antitumor effects was influenced by the viscosity of the culture medium. As viscosity escalated, the results indicated an augmentation in the inhibition of MTT-based viability for PANC1 and PAN198 cells when exposed to LIV-treated lymphocyte-derived CM (Figure 8A,B), as well as for MDA-MA-231 and MDA-MB-436 cells (Appendix A). Remarkably, a consistent enhancement was noted in the suppression of scratch-based motility for PANC1 cells (Figure 8C).

## 4. Discussion

In this study, we demonstrated the inhibitory effects of ALDOA-derived active compound, ACP P04, on the viability, proliferation, migration, and invasion of human PANC1 and PAN198 cells. Additionally, P04 exhibited the ability to reduce the volume of PDAC tissue fragments obtained from patients. Notably, when combined with the chemotherapeutic agents 5-Flu and gemcitabine, P04 demonstrated an additive effect, suggesting the potential for a reduction in the dosage of current chemotherapy drugs. Moreover, the utilization of LIV transformed PBMCs and MSCs into iTSCs. These iTSC-derived CM demonstrated the inhibition of PANC1 cell proliferation and a reduction in the size of PDAC tissue fragments. The proposed mechanism of P04’s action is depicted in Figure 8D. The administration of P04 was correlated with the downregulation of p-Src and snail proteins, while also triggering upregulation of the cytotoxic form of Caspase 3. Of significance, intracellular ALDOA expression within PDAC cells was prominently oncogenic, whereas the application of extracellular rhALDOA protein acted as an inhibitor of PDAC cell growth. Furthermore, our investigation revealed that the overexpression of ALDOA in both MSCs and PBMCs facilitated their conversion into iTSCs. Remarkably, the CM derived from these iTSCs significantly hindered the viability of PDAC cells. Moreover, the application of a technique known as LIV was proven effective in generating iTSCs. LIV not only upregulated ALDOA expression in PBMCs but also endowed their CM with potent tumor-suppressing properties. Noteworthy, the heightened viscosity of the culture medium further amplified the anti-tumor potential of LIV-treated PBMC-derived CM.

P04 shows promise in inhibiting PDAC cells without affecting MSC growth, indicating its tumor-selective potential and reduced side effects. P04 treatment on PDAC cells reduces p-Src levels, hindering cancer cell growth and invasion, and slowing tumor progression. It also downregulates Snail, potentially preventing EMT, cancer cell invasion, and metastasis. Most importantly, P04 upregulates c-caspase3, potentially inducing tumor cell apoptosis. The EGFR belongs to the ErbB protein family, including EGFR (ErbB1, HER1), ErbB2 (HER2), ErbB3 (HER3), and ErbB4 (HER4). While they can form heterodimers, their amino acid sequences vary significantly. EGFR interacts with aldolase A via 15 residues, whereas HER2, HER3, and HER4 have only 7, 5, and 5 residues, respectively, for this interaction. Investigating P04’s potential inhibitory effects on other ErbB protein members requires further analysis.

Drawing from our previous investigation of P05, which shares its amino acid sequence origin with ALDOA, it is important to underscore that the anticancer effectiveness of P04 does not rely on electric charges. This approach represents a departure from indiscriminate electrostatic interactions with negatively charged cancer cells. To assess the potential binding of P04 to EGFR, we employed the Z-DOCK protein docking program. Our predictions revealed four feasible interactions within the 2.6 to 3.0 Å range between P04 and EGFR. It is noteworthy that we previously anticipated the prospective binding of EGFR to P05, the alternate ACP derived from ALDOA. These findings shed light on a downstream signaling pathway associated with EGFR [37], which encompasses K-Ras, a small GTPase in the receptor tyrosine kinase family [38]. The findings indicated a discernible reduction in K-Ras expression within PANC1 cells upon the application of P04. Given the prevalent occurrence of aberrant K-Ras activation among PDAC patients, which significantly influences clinical prognoses, these outcomes hold substantial implications. Collectively, this study highlights the possible involvement of EGFR in mediating the anti-PDAC effects of P04, consequently leading to the inhibition of K-Ras signaling.

Compared to other ACP studies, ACP candidates in this study are unique since they were derived from iTSC-secreted tumor-suppressing proteins. For example, to prevent tumor-linked angiogenesis, the VEGFR1 epitope peptide (SYGVLLWEI) [39] and the VEGFR2 epitope peptide (RFVPDGNRI) [40] were developed. However, anti-VEGF therapy alone cannot usually cure PDAC. P04 can be used in combination with chemotherapeutic agents. The isoelectric point of P04 is 5.3, in comparison to 10.95 for VEGF1/2 peptides. P04 is closer to the isoelectric point of pancreatic tissue and is slightly acidic, suggesting superior solubility, stability, and biological activity. An ACP prediction software, iDCP-Dose2.1, gave a poor score of 0.039 to P04, and 0.154 and 0.34 to VEGF1/2 targeted peptides, in which the score is in the range of 0 (poorest) to 1 (highest). Collectively, further studies are necessary to predict proper ACPs for the treatment of PDAC.

The concept of generating iTSCs through the overexpression of ALDOA presents an intriguing paradox, considering that ALDOA, a glycolytic enzyme, exhibits heightened expression in diverse tumor types, often being linked to tumorigenesis and the advancement of the disease [41]. However, the counterintuitive approach of inducing tumor-suppressive CM is a unique feature of iTSCs, and we have shown that the overexpression of oncogenes such as cMyc and K-Ras converts non-tumor cells such as MSCs and PBMCs into iTSCs. Consistently, intracellular ALDOA acts as oncogenic, while extracellular ALDOA behaves as anti-oncogenic. Compared to cell-based therapies, CM derived from MSCs and PBMCs is anticipated to present low immunogenicity, although it is preferable to use autologous MSCs and PBMCs.

The application of LIV facilitated the conversion of PBMCs into iTSCs, underscoring the significance of the external mechanical milieu in tumor suppression. Recent reports have highlighted the role of LIV in impeding tumor cell proliferation by disrupting cell cycle signaling pathways [42]. Furthermore, LIV is known to promote the activity of immune cells [43]. In this study, we generated iTSCs from PBMCs harvested from healthy individuals as well as patients with PDAC. The use of patient-derived PBMCs should lower the risk of immune rejection. Interestingly, increasing the viscosity of the culture medium enhanced the anti-tumor effect of LIV-treated iTSC CM. The high viscosity may elevate the sensitivity of PBMCs to LIV by altering the mechanical properties of PBMCs, such as membrane rigidity, and stiffness of the cytoskeleton [44,45]. Viscosity may also affect cell morphology and cell–matrix interactions [46], as well as the synthesis and secretion of proteins [47]. Overall, LIV is a useful procedure to generate iTSCs without using any chemical or biological agents.

PBMCs can be procured from individuals with PDAC without necessitating surgical intervention. In this investigation, we transformed PBMCs sourced from PDAC patients into iTSCs by either overexpressing ALDOA or employing LIV techniques. The inherent capacity of iTSC CM to counteract tumors was demonstrated using PDAC tissue fragments derived from the same patient cohort. Moreover, the synergistic integration of P04 with conventional chemotherapeutic agents, along with the administration of iTSC CM, markedly augmented the anti-tumor efficacy while concurrently lowering the IC_50_ of the chemotherapeutic compounds. This combined strategy exhibits the potential to impede tumor progression through the modulation of various signaling pathways, targeting distinct molecular factors, thereby engendering a potent therapeutic outcome. Cumulatively, the utilization of patient-derived PBMCs as a foundation for therapeutic interventions holds considerable promise. Further rigorous investigations are imperative to assess the safety, effectiveness, and compatibility of both P04 and iTSC CM in conjunction with established treatment modalities.

In contrast to alternative biological treatments, ACPs offer notable advantages, encompassing synthetic cost-effectiveness and stringent quality control measures. Nevertheless, ACPs carry potential drawbacks, concerning their stability, tissue penetrability, and in vivo cytotoxicity. Concerns include potential side effects on surrounding tissues and the emergence of resistance. It is important to acknowledge that the scope of our present study is constrained. A thorough investigation remains imperative to elucidate the stability, specificity, and potential side effects inherent to P04. Although molecular docking studies have hinted at a possible interaction between P04 and EGFR, the definitive mechanistic underpinnings of P04’s anti-PDAC efficacy must be ascertained through experimental validation. Furthermore, the effectiveness of P04 might exhibit variation among patients, encompassing the intricate heterogeneity of PDAC cells such as K-Ras mutations and the state of immune systems, among other factors.

## 5. Conclusions

This study unequivocally establishes P04 as a potent contender for anti-PANC anticancer peptides, effectively curbing the viability, motility, and invasive potential of three distinct PDAC cell lines. P04 seamlessly integrates with prevailing chemotherapeutic agents, substantially diminishing the IC_50_ values associated with each of the three agents under scrutiny. Moreover, this study showcased the generation of iTSCs and the production of tumor-suppressive CM through two distinct approaches: ALDOA overexpression and the utilization of LIV. While additional research and clinical trials are necessary to validate its effectiveness and safety, the development of P04 alongside a distinctive biophysical technique for iTSC generation introduces fresh outlooks and potential avenues in the field of cancer therapy.

## Figures and Tables

**Figure 1 pharmaceutics-15-02447-f001:**
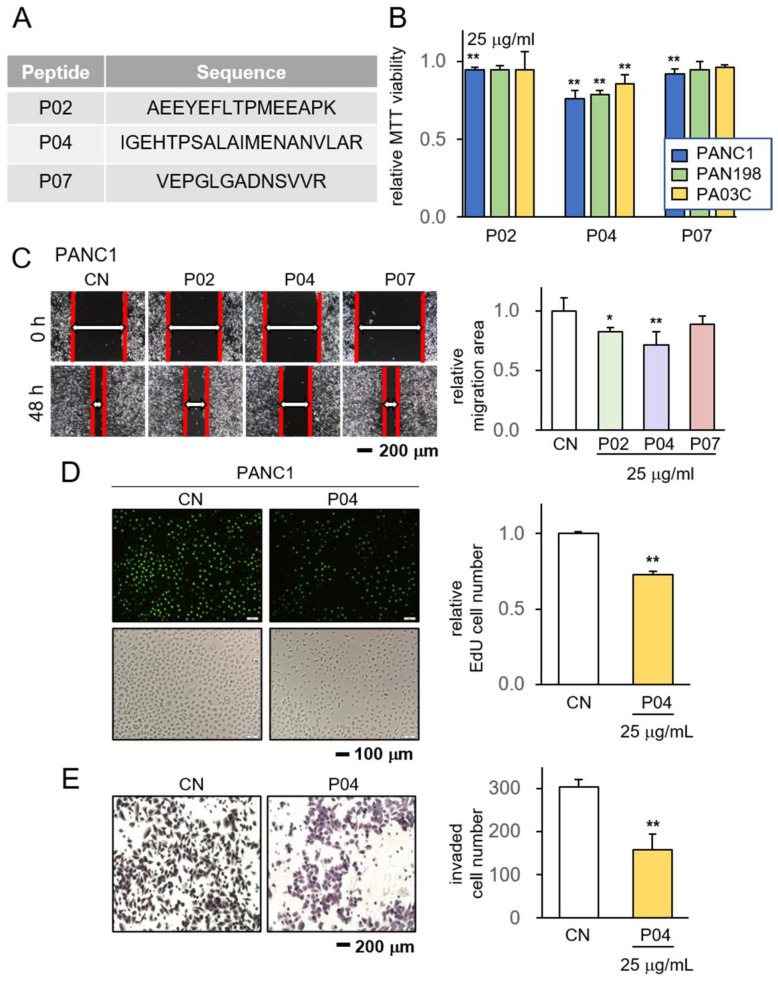
Anti-tumor effect of P04 on PDAC cell lines. CN = control. The single and double asterisks indicate *p* < 0.05 and 0.01, respectively. (**A**) The amino acid sequences of the selected three peptides, including P02, P04, and P07; (**B**) reduction in MTT-based viability of PANC1, PAN198, and PA03C pancreatic tumor cells in response to 25 μg/mL of P02, P04, and P07; (**C**) reduction in scratch-based motility of PANC1 cells in response to 25 μg/mL of P04; (**D**,**E**) suppression of EdU-based proliferation and transwell-based invasion of PANC1 cells in response to 25 μg of P04, respectively.

**Figure 2 pharmaceutics-15-02447-f002:**
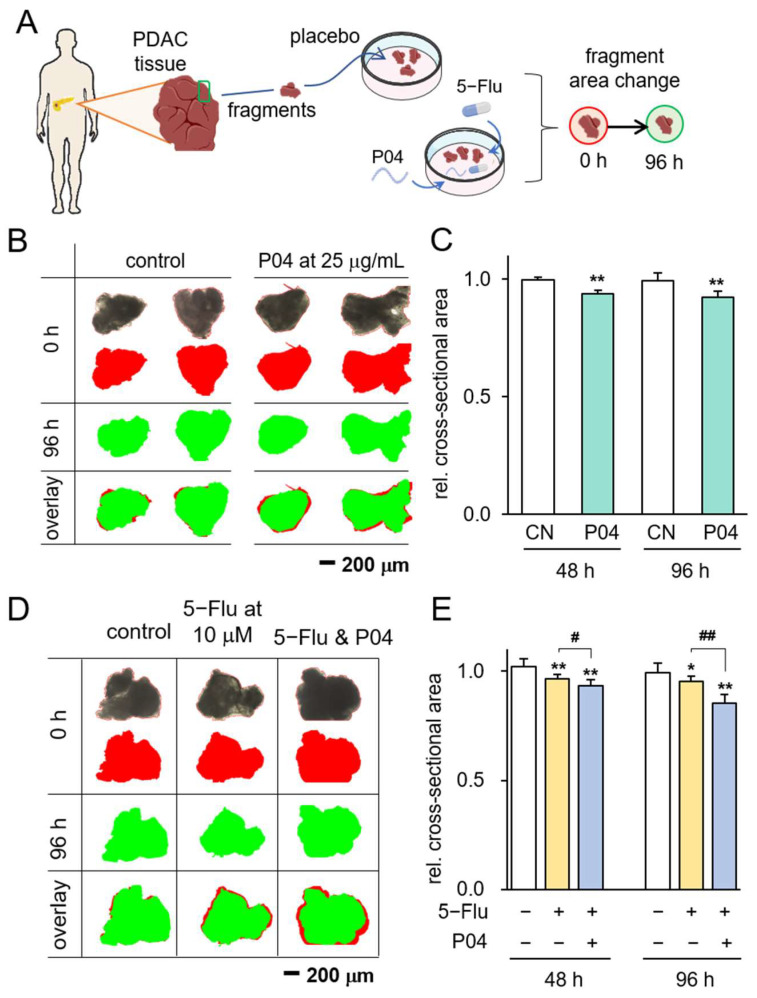
Inhibitory effects on PDAC tissue fragments in response to P04. The single and double asterisks indicate *p* < 0.05 and 0.01 vs. CN, respectively. The single and double hashtag indicate *p* < 0.05 and 0.01, 5-Flu group vs. 5-Flu&P04 group, respectively (**A**) The illustration of PDAC tissue fragments assay; (**B**,**C**) reduction in PDAC fragment size in 96 h in response to 25 μg/mL of P04 (*n* = 8). The red image indicates tissue fragments at 0 h, while the green image indicates them at 96 h; (**D**,**E**) reduction in PDAC fragment size in 96 h in response to 10 μM of 5-Flu and the combination with 25 μg/mL P04 is more effective (*n* = 8). The red image indicates tissue fragments at 0 h, which represents the area of the black original image, while the green image indicates them at 96 h.

**Figure 3 pharmaceutics-15-02447-f003:**
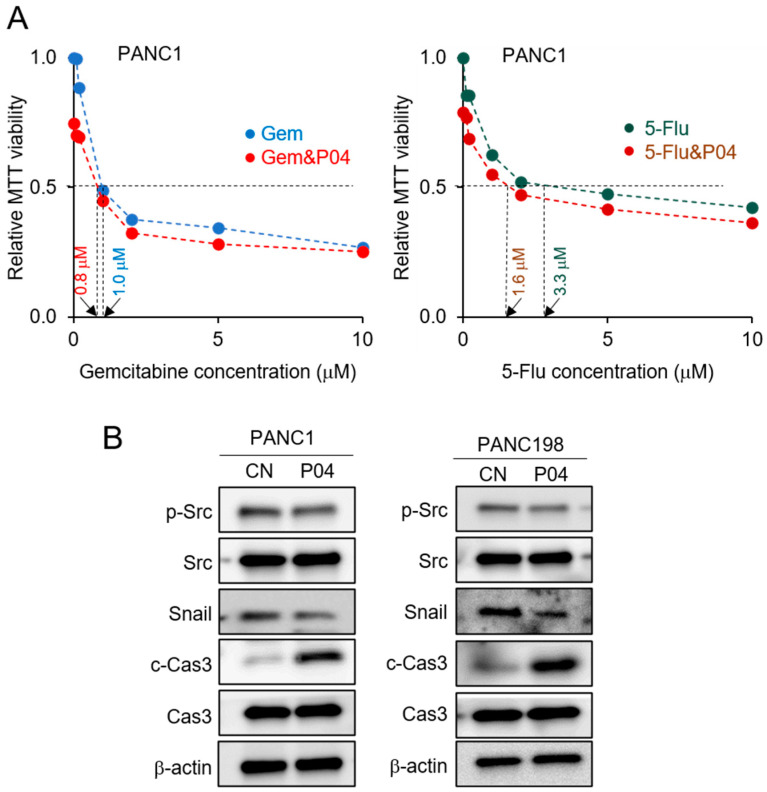
Inhibitory effects of P04 in combination with gemcitabine and 5-Flu. CN = control, Gem = gemcitabine, and 5-Flu = 5-fluorouracil. (**A**) Additive anti-tumor effects of P04, together with gemcitabine or 5-Flu; (**B**) decrease in the levels of p-Src and Snail, as well as an increase in cleaved caspase 3 (c-Cas-3) in PANC1 and PAN198 PDAC cells in response to 25 μg/mL of P04.

**Figure 4 pharmaceutics-15-02447-f004:**
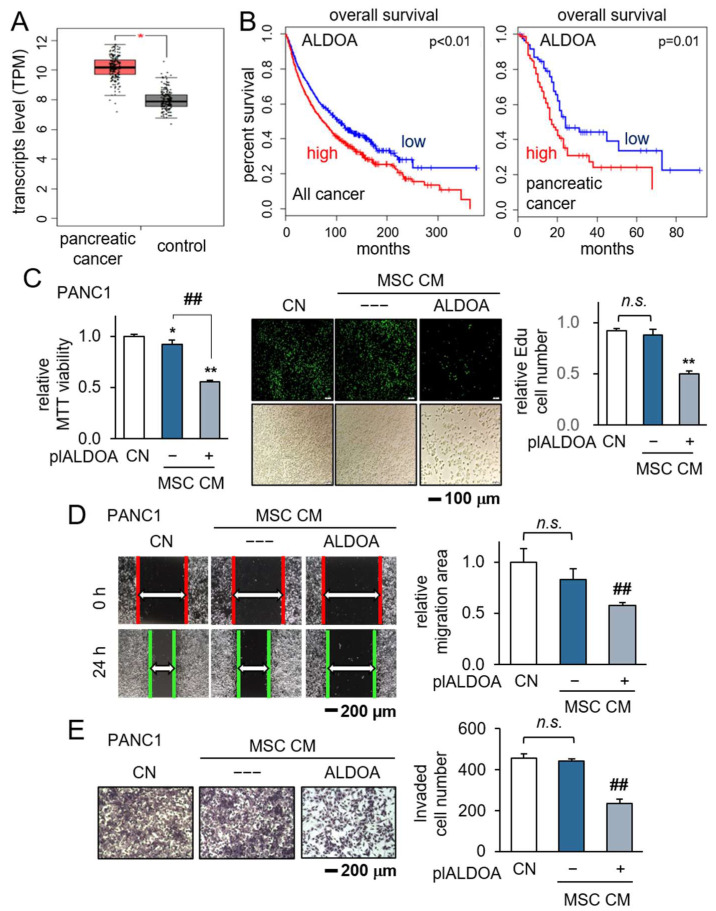
Generation of iTSCs by the overexpression of ALDOA in MSCs. CN = control, pl = plasmid transfection, MSC = mesenchymal stem cell, and CM = conditioned medium. The single and double asterisks indicate *p* < 0.05 and 0.01, MSC CM group vs. CN. The double hashtag indicates *p* < 0.01, MSC-plALDOA CM group vs. MSC-NC CM group, respectively. n.s. = non-significant. (**A**,**B**) The elevated ALDOA transcript level in patients with PDAC, and lower survival rate for patients with a high ALDOA level in TCGA database; (**C**–**E**) reduction in MTT-based viability, EdU-based proliferation, scratch-based motility, and transwell invasion, respectively, by ALDOA-overexpressing MSC-derived conditioned medium.

**Figure 5 pharmaceutics-15-02447-f005:**
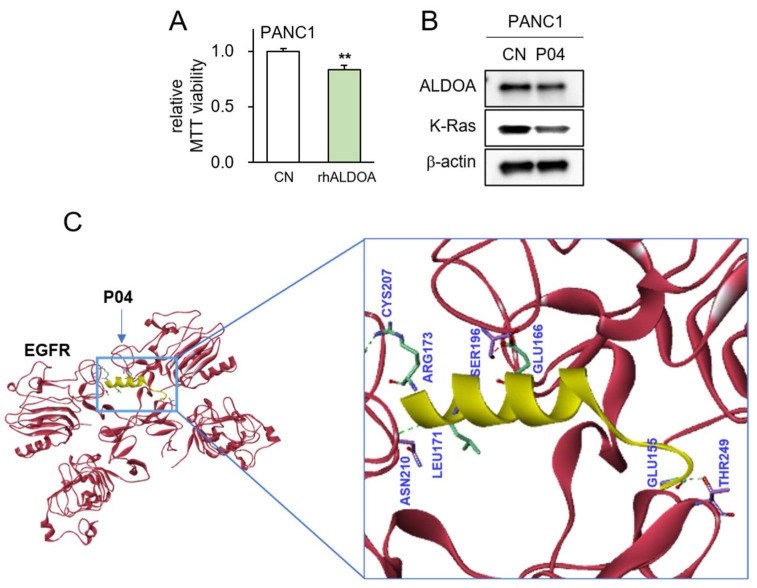
Predicted interaction between P04 and EGFR. The double asterisk indicates *p* < 0.01 vs. CN (**A**) Reduction in MTT-based viability of PANC1 cells in response to 5 μg/mL of recombinant ALDOA protein; (**B**) decrease in the levels of ALDOA and K-Ras in PANC1 cells by the application of 25 μg/mL of P04; (**C**) predicted interactions between P04 and EGFR by molecular docking using Z-DOCK software 2016.

**Figure 6 pharmaceutics-15-02447-f006:**
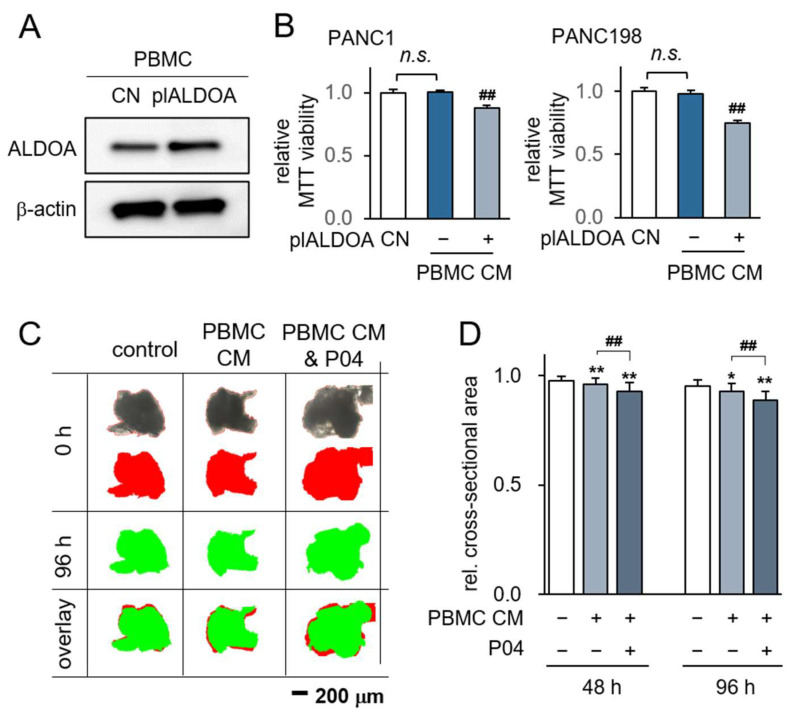
Generation of iTSCs from PBMCs by overexpressing ALDOA. CN = control, pl = plasmid transfection, PBMC = peripheral blood mononuclear cells, and CM = conditioned medium. (**A**) Elevated ALDOA level by transfection of ALDOA plasmids; (**B**) reduction in MTT-based viability of PANC1 and PAN198 cells by ALDOA-overexpressing PBMC-derived CM. The double hashtag indicates *p* < 0.01, PBMC-plALDOA CM group vs. PBMC-NC CM group, respectively. n.s. = non-significant; (**C**,**D**) shrinkage of PDAC fragments by ALDOA-overexpressing PBMC-derived CM with and without P04. The red image indicates tissue fragments at 0 h, which represents the area of the black original image, while the green image indicates them at 96 h. The single and double asterisk indicates *p* < 0.05 and 0.01 vs. CN, respectively. The double hashtag indicates *p* < 0.01, PBMC CM & P04 group vs. PBMC CM group.

**Figure 7 pharmaceutics-15-02447-f007:**
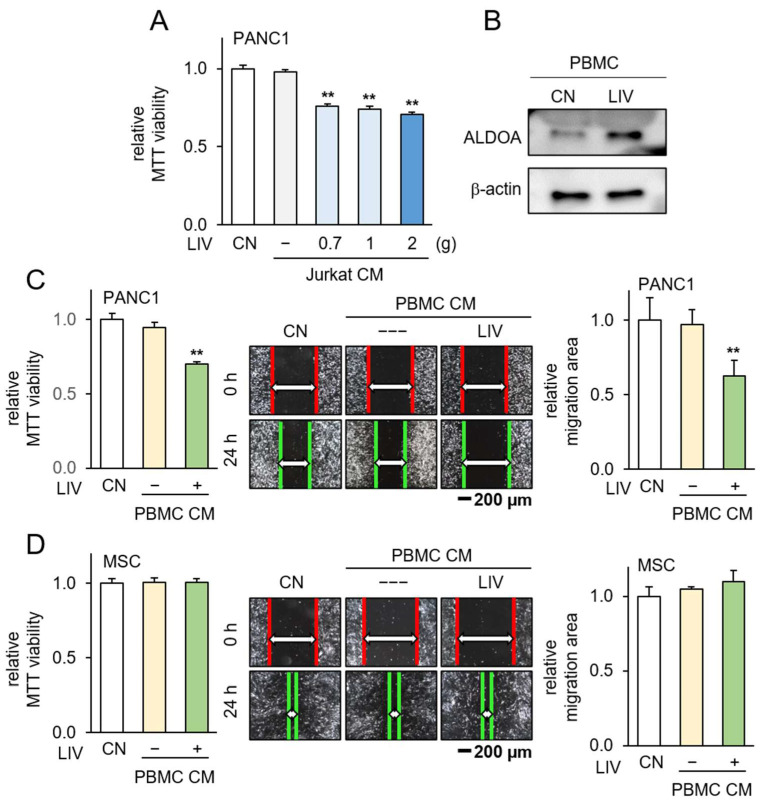
Generation of iTSCs from PBMCs by the application of low-intensity vibration. CM = conditioned medium, CN = control, and LIV = vertical low-intensity vibration. The double asterisk indicates *p* < 0.01 vs. non-LIV treated group. PBMC CM & P04 group vs. PBMC CM group. (**A**) Reduction in MTT-based viability of PANC1 cells by the application of LIV-treated Jurkat cell-derived CM; (**B**) an elevated level of ALDOA in PBMCs in response to LIV; (**C**) reduction in MTT-based viability and scratch-based motility of PANC1 cells by LIV-treated PBMC-derived CM; (**D**) no detectable changes in MTT-based viability and scratch-based motility of MSCs by LIV-treated PBMC-derived CM.

**Figure 8 pharmaceutics-15-02447-f008:**
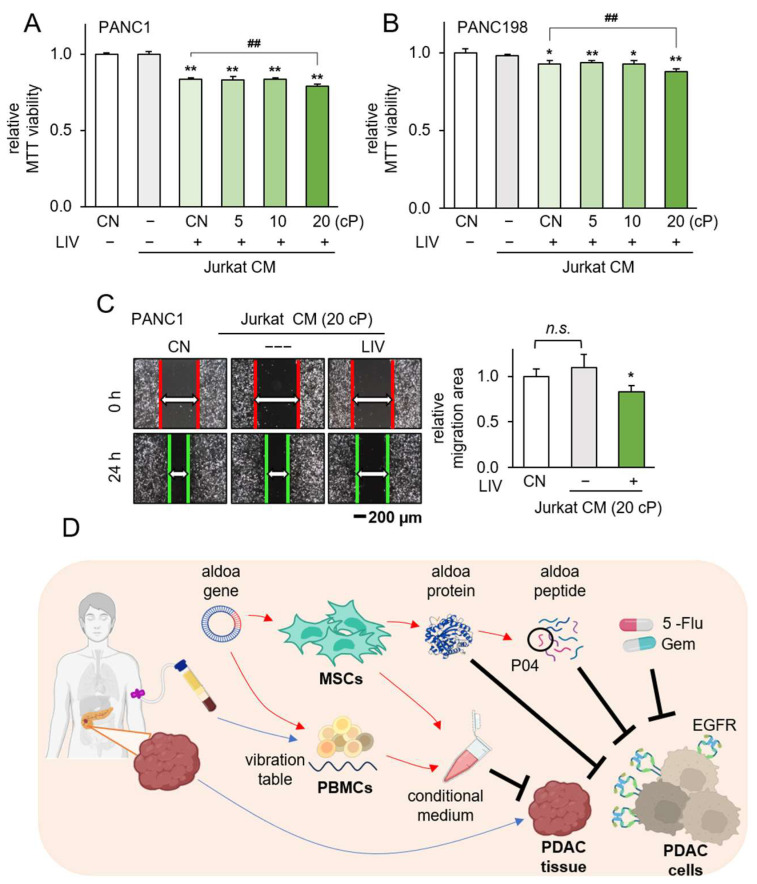
Generation of iTSCs from Jurkat cells by LIV at different viscosity. CM = conditioned medium, CN = control, LIV = vertical low-intensity vibration, and cP = centipoise (viscosity unit). (**A**,**B**) Reduction in MTT-based viability of PANC1 and PAN198 cells by the application of LIV-treated Jurkat-derived CM. The single and double asterisk indicates *p* < 0.05 and 0.01, respectively vs. non-LIV treated Jurkat CM group. The double hashtag indicates *p* < 0.01, Jurkat LIV treated at 20 cP CM group & Jurkat LIV treated at normal medium CM group. n.s. = non-significant; (**C**) reduction in scratch-based motility of PANC1 cells by LIV-treated Jurkat-derived CM; (**D**) proposed mechanism of the anti-tumor action of ALDOA, P04, and vibration-treated PBMC-derived CM.

## Data Availability

The data presented in this study are available on request from the corresponding author.

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
