# Peer review of "Anticancer Peptides Derived from Aldolase A and Induced Tumor-Suppressing Cells Inhibit Pancreatic Ductal Adenocarcinoma Cells"

_pharmaceutics, 2023, doi:10.3390/pharmaceutics15102447_

Round 1
Reviewer 1 Report
The study by Ciu et al. is focused on the identification and characterization of the novel peptides from induced tumor-suppressing cells for inhibiting pancreatic cancer cells. Thus the authors report that peptide P04 serves as a potent contender for anti-PANC ACP, effectively curbing the viability, motility, and invasive potential of three distinct PDAC cell lines. These results are novel, while the study is well designed, conclusions are supported by the presented data, the manuscript is well written and easy to follow. Therefore, I am happy to recommend it for the publication. The only thig I would like to suggest the authors is thinking about the title. The current version is too broad.
Reviewer 2 Report
-The current work focuses on anticancer peptides from induced tumor-suppressing cells for inhibiting pancreatic cancer cells. The author’s great effort into the manuscript, but minor issues should be addressed.
Major:
-The introduction provides sufficient background and all relevant references are included, but the novelty of this work is not highlighted and the author's contribution was unclear compared to other previous works.
-One of the main problems in the manuscript is that the authors show only results without interpretations of details or confirmation by citation. More details are required to explain the obtained results.
e.g. Line 217-219. “P04 stood out as manifesting the most potent anti-tumor effects, particularly evident in its -capacity to curtail MTT-based cell viability and hinder scratch-based cell motility within PANC1 cells” Why? what is the advantage and expected mechanism for P04 than others? and need citations to support this claim.
Line 226-228. “P04 exhibited a discernible decrease in the proliferation and invasive characteristics of PANC1 cells” What is the role and mechanism? Citation?
Line 244-246. “Furthermore, our observations indicated that the combined application of P04 and 5-Flu exhibited superior efficacy compared to the usage of 5-Flu alone, resulting in a more substantial reduction in the size of PDAC fragments” mechanism, Citation?
In Fig. 8 d, “Proposed mechanism of the anti-tumor action of ALDOA, P04, and vibration treated PBMC-derived CM.” the authors show only figures without any comments on it and also not mentioned in the text.
Minor:
-Figures need higher resolution and appearance for easier reading by the reader.
-In conclusion "It's essential to underscore that P04's anticancer efficacy is rooted in targeted mechanisms
and not reliant on indiscriminate electrostatic interactions with negatively charged cancer
cells.". before we can say this, it should be at least one paragraph in result and discussion to discuss this point in detail.
Reviewer 3 Report
In this study, the authors explored a unique approach to identify potential anticancer peptides (ACPs) for PDAC treatment. They utilized 20 previously developed induced tumor-suppressing cells (iTSCs) known to secrete a group of tumor-suppressing proteins. The research focused on identifying ACPs within trypsin-digested protein fragments derived from iTSCs. Among the 10 ACP candidates examined, peptide P04 (IGEHTPSLAIMENANVLAR) showed the most efficient anti-PDAC activities. Importantly, P04 was found to be derived from Aldolase A (ALDOA), a glycolytic enzyme. Further investigations revealed that extracellular ALDOA, as well as P04, were predicted to interact with the epidermal growth factor receptor (EGFR). P04 also exhibited the ability to downregulate oncoproteins such as Snail and Src, specifically targeting PDAC cells without affecting mesenchymal stem cells (MSCs). To enhance the therapeutic potential of P04, the researchers generated iTSCs by overexpressing ALDOA in both MSCs and peripheral blood mononuclear cells (PBMCs). The conditioned medium (CM) derived from these iTSCs demonstrated inhibitory effects on PDAC cell progression and tissue fragments. Importantly, the inhibitory effect of P04 was additive when combined with CM and conventional chemotherapeutic drugs like 5-Flu and Gemcitabine. Intriguingly, the study also revealed a unique approach to convert PBMCs into iTSCs by applying mechanical vibration, leading to elevated ALDOA levels and phosphorylated Akt in the PI3K signaling pathway.
Overall, this is a very interesting study which is well designed, performed and presented. I would recommend the publication of this study after addressing the following concerns:
- the authors should extend the intro part behind the molecular mechanisms of ACPs, and related targeted proteins... there are a major piece of data there to be covered and highlighted.
- in the material and M section, there are several techniques stated without any relevant citation to the applied protocol... e.g.MTT, scratch assay, transwell invasion..etc..
- regarding the molecular modelling, the authors should add pdb code for EGFR ustilized. More details for the docking protocol and applied approach should be added (pls refer to doi.org/10.3390/pharmaceutics14030529, doi.org/10.4155/fmc-2021-0066, doi.org/10.3390/ph15070832)
- regarding the MTT screening, why the authors have not add a positive control drug? why not assessing dose-dependent, IC50? why 25ug/mL?
- for TableS2, at which conc.??
- what is the effect IC50 of PC4 on normal cell viability?
- as observed in Fig3, the synergistic effect of P4 with anticancer drug is not that substantial, how the authors can explain this? pls add SD/SE for fig3, was this assessed in triplicate?
- the molecular modelling study is not well presented. The authors should present the results in more details, what is crystal structure used? what is the essential amino acids in binding site? what is the control ligand utilized ...
- why the authors have not assessed by flow cytometry analysis the cell cycle to gain more insights about P04.
- please extend the conclusion more to highlight the future prospective for the identified probe.
- just curious, do the authors think that peptide-based probes would be stable in cells? could be off-targeted and hydrolyzed?
minor editing would be required.
Reviewer 4 Report
Comments attached

Round 2
Reviewer 3 Report
the authors have adequately addressed the raised concerns.
minor editing is requested.